# Trend and Geographic Disparities in the Mortality Rates of Primary Systemic Vasculitis in the United States from 1999 to 2019: A Population-Based Study

**DOI:** 10.3390/jcm10081759

**Published:** 2021-04-18

**Authors:** Alicia Rodriguez-Pla, Jose Rossello-Urgell

**Affiliations:** 1Division of Rheumatology, Mayo Clinic Arizona, Scottsdale, AZ 85259, USA; 2Statistics and Epidemiology Consultant, ARJR Media LLC, Scottsdale, AZ 85259, USA; editor@arjrmedia.com

**Keywords:** vasculitis, mortality, giant cell arteritis, ANCA-associated vasculitis, granulomatosis with polyangiitis, microscopic polyangiitis, eosinophilic granulomatosis with polyangiitis, antiglomerular basal membrane disease

## Abstract

The current data on rates and geographic distribution of vasculitis mortality are limited. We aimed to estimate the mortality rates of primary systemic vasculitis and its geographic distribution using recent population data in the United States. The mortality rates of vasculitis from 1999 to 2019 were obtained from the Center for Disease Control (CDC) Wonder Multiple Cause of Death (MCD). The age-adjusted rates per million for vasculitis as MCD and as an underlying cause of death (UCD) were calculated by state using demographics. A joinpoint regression analysis was applied to evaluate trends over time. The age-adjusted mortality rate of vasculitis as MCD was 4.077 (95% CI: 4.029–4.125) and as a UCD was 1.888 per million (95% CI: 1.855–1.921). Since 1999, mortality rates have progressively decreased. The age-adjusted mortality rate was higher in females than in males. The highest mortality rate for vasculitis as MCD was in White patients (4.371; 95% CI: 4.317–4.424). The northern states and areas with lower populations had higher mortality rates. We found a trend of progressive decreases in the mortality rates of vasculitis, as well as gender, racial, and geographic disparities. Further analyses are warranted to better understand the factors associated with these disparities in order to implement targeted public health interventions to decrease them.

## 1. Introduction

Primary systemic vasculitides are a group of heterogeneous disorders characterized by inflammation of the vessel wall. According to the 2012 International Chapel Hill Consensus Conference on the Nomenclature of Vasculitides, vasculitides are classified by the size of the vessel primarily involved in large-, medium-, and small-vessel vasculitis [1]. Large-vessel vasculitis primarily affects the aorta and supra-aortic branches. Giant cell arteritis (GCA) and Takayasu arteritis (TAK) are the two major variants of large-vessel vasculitis. Medium-vessel vasculitis primarily affects the main visceral arteries and their branches. Polyarteritis nodosa (PAN) and Kawasaki disease are the major variants. Small-vessel vasculitis predominantly affects the intraparenchymal arteries, arterioles, capillaries, and venules. The two major categories of small-vessel vasculitis are the subgroup of antineutrophil cytoplasm antibody (ANCA)-associated vasculitis (AAV), which includes granulomatosis with polyangiitis (GPA), eosinophilic granulomatosis with polyangiitis (EGPA), and microscopic polyangiitis (MPA), and the immune complex small-vessel vasculitis, which includes antiglomerular basement membrane (anti-GBM) disease, cryoglobulinemic vasculitis, IgA vasculitis (IgAV) and hypocomplementemic urticarial vasculitis. Additional categories are variable vessel vasculitis, which includes Behcet’s disease, single-organ vasculitis, and vasculitis associated with systemic disease or probable etiology [1].

Earlier diagnosis and less toxic immunosuppressive therapies have improved the survival of patients with vasculitis. Without treatment, primary systemic vasculitis is associated with high mortality rates, especially in patients with AAV, a subtype of small-vessel vasculitis [2] that includes GPA, EGPA, and MPA, as mentioned above [1]. Before treatment with corticosteroids and cytotoxic agents, patients with GPA had a one-year mortality rate between 50% and 80% [3]. During the 1960s, the prognosis improved with the use of cytotoxic agents [4]. In the 1970s, combination therapy with cyclophosphamide plus high-dose steroids for three to six months, followed by less toxic regimens with azathioprine and low-dose steroids, achieved an 80–90% survival rate for AAV [5,6]. It is unknown if regimens with rituximab have further improved the survival for AAV [7].

In medium-vessel vasculitis, better hepatitis B infection treatments have improved the outcomes of PAN [8]. Increased awareness and earlier diagnosis of Kawasaki disease have resulted in earlier treatment and better outcomes and survival [9]. Increased awareness of the need for a more aggressive course in some patients with large-vessel vasculitis with a higher risk of aneurysm formation and ischemic complications [10,11] may have decreased the morbidity and mortality of GCA. It is unknown if interleukin (IL)-6 inhibition has impacted mortality in GCA [12].

The geographic distribution of mortality in different diseases, including stroke [11], heart disease [13], and scleroderma [14], has been reported in epidemiological studies. Environmental factors, sociodemographic characteristics, lifestyle variables, and access to care have been postulated to be important contributors [11].

The current data on the mortality rates of vasculitis and their geographic distribution are limited. In this study, we aimed to estimate the mortality rates and the geographic distribution of the mortality of primary systemic vasculitis using the most recent available population mortality data in the United States. We found a trend of a progressive decrease in the mortality rates of vasculitis, as well as gender, racial, and geographic disparities.

## 2. Materials and Methods

We used the Center for Disease Control (CDC) Wonder database and its query system to obtain the mortality rates of vasculitis as multiple cause of death (MCD) [15] or as the underlying cause of death (UCD) [16] from 1999 to 2019.

The mortality data were based on the information included in all death certificates filed in the 50 states and the District of Columbia. Fetal deaths and deaths of nonresidents and residents of other U.S. territories were excluded. The causes of death were classified in accordance with the International Classification of Disease (ICD). Deaths for 1999 and beyond were classified using the Tenth Revision (ICD-10) [15].

The MCD data available in the CDC Wonder database are country-level national mortality and population data from 1999 to 2019. Each death certificate identifies a single UCD, plus up to 19 additional causes and demographic data. The World Health Organization (WHO) defined a UCD as “the disease or injury which initiated the train of events leading directly to death, or the circumstances of the accident or violence which produced the fatal injury.” The UCD is selected from the conditions entered by the physician in the cause of death section of the death certificate. When more than one cause is entered, it is determined by the sequence of conditions on the certificate, the provision of the ICD, and the associated selection rules and modifications [15]. When one of the primary vasculitides is listed as the UCD, it means that vasculitis was the disease that initiated the event leading directly to death. When one of the primary vasculitides is considered a contributing cause, but not the UCD, it is listed among the 19 additional causes of death on the certificate. The patients can be represented more than once in the MCD statistics if they have more than one vasculitis.

As stated on the CDC Wonder website, race and Hispanic origin are reported separately on the death certificate. The American Indian or Alaska Native race category includes North, Central, and South American Indians, Eskimos, and Aleuts. The Asian or Pacific Islander race category includes Chinese, Filipino, Hawaiian, Japanese, and Other Asian or Pacific Islanders. As Hispanic origin is not reported on the death certificate for some deaths, missing Hispanic origin information is coded as “not stated” and these deaths are excluded when death rates are calculated based on Hispanic origin [15].

To query the CDC Wonder database, we used the following ICD-10 codes: D69.0 (allergic purpura) for IgAV (Henoch–Schönlein), D89.1 (cryoglobulinemia) for cryoglobulinemia, M30.0 (polyarteritis nodosa) for PAN, M30.1 (polyarteritis with lung involvement (Churg–Strauss)) for EGPA, M30.2 (juvenile polyarteritis) for juvenile polyarteritis, M30.3 (mucocutaneous lymph node syndrome (Kawasaki)) for Kawasaki disease, M30.8 (other conditions related to PAN) for the same, M31.0 (hypersensitivity angiitis) for antiglomerular basement membrane (anti-GBM) disease, M31.3 (Wegener’s granulomatosis) for GPA, M31.4 (aortic arch syndrome (Takayasu)) for TAK, M31.5 (giant cell arteritis with polymyalgia rheumatica) for GCA with polymyalgia rheumatica (PMR), M31.6 (other giant-cell arteritis) for GCA, M31.7 (microscopic polyangiitis) for MPA, and M35.2 (Behcet’s disease) for Behcet’s disease.

Mortality rates were obtained by year, gender, race, ethnicity, state, and the urban–rural continuum according to the 2013 urbanization area classification. To obtain age-adjusted mortality rates, we used a U.S. standard population of year 2000. Mortality rates are given as number of deaths per million. A joinpoint regression analysis was used to calculate the average annual percent change from 1999 to 2019. This analysis is also able to identify changes in trends over time [17].

## 3. Results

### 3.1. Overall Age-Adjusted Mortality Rates

During the 21-year period of our study, based on a cumulative population of 6,416,872,524 individuals, vasculitis was reported among the MCD in 28,190 patients. Out of them, vasculitis was reported as the UCD in 13,048 patients (46.29%). The age-adjusted mortality rate of vasculitis as MCD was 4.077 per million (95% CI: 4.029–4.125) and as the UCD was 1.888 per million (95% CI: 1.855–1.921). Since 1999, there has been a significant decreasing trend for vasculitis, both as MCD with an average annual percent change (AAPC) of −3.7% (95% CI: −4.5% to −2.8%) and as the UCD with an AAPC of −3.8% (95% CI: −5% to −2.5%) (Figure 1). For MCD, the joinpoint analysis identified three different time periods with their respective annual percent changes (APCs): 1999–2008, APC −4.3% (95% CI: −5.1% to −3.5%); 2008–2016, APC −1.8% (95% CI: −3% to −0.5%); 2016–2019, APC −6.6% (95% CI: −11.1% to −1.7%). For the UCD, two time periods were identified: 1999–2016, APC −2.8% (95% CI: −3.3% to −2.2%); 2016–2019, APC −9.3% (95% CI: −5% to −2.5%).

### 3.2. Mortality Rates by Gender

An asterisk indicates that the Annual Percept Change (APC) is significantly different from zero at the alpha = 0.05 level. 

Considering gender, the percentage of vasculitis deaths was higher in females than in males both for vasculitis as MCD (58.95% vs. 41.05%) and for vasculitis as the UCD (55.50% vs. 44.51%). The age-adjusted mortality rate was also higher in females than in males for vasculitis as MCD (5.057 vs. 3.641 per million) and as the UCD (2.220 vs. 1.841 per million).

### 3.3. Mortality Rates by Race

Per race, most deaths occurred in Whites (92.34%) for vasculitis as MCD. Whites also had the highest mortality rate (4.371; 95%CI: 4.317–4.424), followed by American Indians or Alaska Natives (3.451; 95% CI: 2.910–3.992), Blacks (2.037; 95%CI: 1.929–2.146) and finally Asians or Pacific Islanders (1.861; 95% CI: 1.696–2.026). For UCD, the highest percentage of total vasculitis deaths also occurred in Whites (91.36%), but the mortality rate was slightly higher in American Indians or Alaska Natives (2.059; 1.643–2.475) than in Whites (2.032; 1.995–2.069) (Table 1). In Whites, vasculitis was the UCD in 46.13% of all vasculitis deaths. This percentage was the highest in American Indians or Alaska Natives (60.44%), followed by Asians or Pacific Islanders (52.50%) and then Blacks (51.66%).

### 3.4. Mortality Rates by the Hispanic or Non-Hispanic or Latino Groups

Interestingly, 94.06% of the deaths due to vasculitis as MCD and 92.91% of those as the UCD happened in the non-Hispanic or Latino groups. The mortality rates were 2.819 (95% CI: 2.673–2.964) for Hispanic or Latino vs. 4.136 (95% CI: 4.086–4.186) for non-Hispanic or Latino for vasculitis as MCD and 1.079 (95% CI: 1.414–1.625) for Hispanic or Latino vs. 1.945 (95% CI: 1.910–1.980) for non-Hispanic or Latino in the case of vasculitis as the UCD.

### 3.5. Mortality Rates by Race and Gender

When cases were stratified by race and gender, for vasculitis as MCD, the highest rate was in White females at 4.469 (95% CI: 4.397–4.542), followed by White males with a rate of 4.176 (95% CI of 4.095–4.256). In the case of the UCD, the highest rates were in American Indian or Alaska Native females at 2.429 (95%CI: 1.867–3.108), followed by White males with a rate of 2.052 (95% CI: 1.996–2.108), and then very closely by White females with a rate of 2.006 (95% CI: 1.957–2.056). The lowest rates were seen in Asian or Pacific Islander males both for vasculitis as MCD and as the UCD (Table 1).

### 3.6. Mortality Rates by Vasculitis Type

When analyzing the percent contribution of each vasculitis to mortality, we found that, as the UCD, GPA was the vasculitis responsible for half of the deaths, accounting for 50.18% of them, followed by GCA (13.97%) and anti-GBM disease (10.07%) (Table 2). The corresponding percentages for vasculitis as MCD were 36.85%, 32.64%, and 6.78%, respectively.

For the three vasculitides contributing the most to the MCD data, we analyzed the frequency of the UCD found on those death certificates. In GPA, GPA itself contributed 63.03% of all UCD, followed by atherosclerotic heart disease (2.29%), acute myocardial infarction (1.85%), and chronic obstructive pulmonary disease (1.54%). In GCA, GCA itself contributed 19.80%, followed by atherosclerotic heart disease (6.77%), stroke (5.54%), and acute myocardial infarction (5.15%). Finally, for anti-GBM disease, the disease itself contributed 68.69%, followed by acute myocardial infarction (1.20%), atherosclerotic heart disease (1.15%), and hypertensive renal disease with renal failure (1.0%).

### 3.7. Geographic Distribution of the Vasculitis Mortality Rates

There was geographic disparity in the mortality rates of vasculitis by states, both as MCD and as the UCD. The states with the highest mortality rates for vasculitis as MCD were Oregon, Vermont, Minnesota, Maine, and Idaho, and as the UCD were Oregon, Maine, Vermont, Alaska, and Wyoming. The states with the lowest rates for vasculitis as MCD were Louisiana, Nevada, District of Columbia, Florida, and New York, and as the UCD were New York, Hawaii, New Jersey, District of Columbia, and Rhode Island.

Looking at a map of the U.S., the states with the highest mortality both for vasculitis as MCD and the UCD were situated in the north of the United States (Figure 2 and Figure 3). In the case of vasculitis as MCD, the north–south differential was very evident, with all of the states with higher mortality rates falling above a horizontal line that could be drawn just south of Oregon, Idaho, Wyoming, Nebraska, Iowa, and New Hampshire (Figure 2). In the case of vasculitis as the UCD, there were only two southern states, New Mexico, and Kansas, that had high mortality rates. In the north, only North Dakota, South Dakota, New York, and a few states in the north east, including Connecticut and Rhode Island, had low mortality rates for vasculitis as the UCD (Figure 3).

### 3.8. Distribution of Vasculitis Mortality Rates According to the Urban–Rural Continuum

When looking at the distribution of vasculitis mortality according to the urban–rural continuum, there was a clear progressive decreasing trend of mortality rate as the size of the population increased both for vasculitis as MCD and as the UCD (Table 3).

## 4. Discussion

We found that during the 21-year period of our study, the age-adjusted mortality rate of vasculitis as MCD was 4.077 and as the UCD was 1.888 per million. Since 1999, there has been a significant decreasing trend for vasculitis mortality both as MCD and as the UCD. Since 2016, the decrease in mortality rate has been more pronounced.

Current data on the population mortality rates of vasculitis are scarce. A systematic review of the mortality in systemic vasculitis in 2008 found that GPA patients had a 75% survival rate at four years. At five years, the survival rate was 45–75% for MPA, 68–100% for EGPA, 70–93% for TAK, 75% in adult-onset IgAV, 75–80% for PAN, greater than 99% for Kawasaki disease, and GCA had a survival rate equivalent to the age-matched population [2]. Using earlier versions of the same database of our current study, we previously reported a progressive decrease in the mortality rates for vasculitis as the UCD in the United States from 1999 to 2010 [18] and from 1999 to 2017 [19]. Population data from the United Kingdom showed a decreased AAV-related mortality over 20 years [20]. In 2017, a meta-analysis of 10 observational studies on the morality of AAV from 1966 to 2009 found a 2.7-fold increased risk of death in patients with AAV and a 2.6-fold increased risk in patients with GPA when compared with the general population. Subgroup analyses between 1980 and 2005 showed that the mortality risk improved over time. A recent study on the mortality of AAV in the U.S. from 1999 to 2017 found that there were 11,316 AAV-related deaths in the U.S., with an age-adjusted mortality rate of 1.86 per million for AAV. The mortality rate was highest among non-Hispanic White people, men, and persons in the Midwest. Overall, the mortality rate decreased from 1997 to 2017 by an average of 1.6% per year [21]. The more pronounced decrease in mortality rates since 2016 that we found in our study is probably the result of a combination of increased physician and patient awareness about these rare disorders, which has resulted in earlier diagnosis and treatment and has improved and lessened toxic therapeutic regimens.

We found that the mortality rates were higher in females than in males for vasculitis both as MCD and as the UCD. A recent meta-analysis found that in five of the 10 studies selected, risks were evaluated separately according to gender, and no difference was seen [22]. This finding contradicts a previous report from Germany, which found that young men with GPA had an 8.87-fold increased risk of death than young women with the disease, and this was thought to be due to more frequent kidney involvement in men [23]. In AAV, the mortality rates were reported to be higher in males than in females [21]. It is well documented that women and men differ in patterns of illness and disease risk factors, but also in social context, and differences in access, quality, and health outcomes have been evidenced. Access to intensive care and to potentially life-saving interventions is lower for critically ill women 50 years and older [24]. Other studies have found women to be at increased risk of receiving suboptimal care for serious illnesses [25]. If our finding of increased mortality in females is confirmed by other studies, additional research to elucidate the causes of these gender disparities in the mortality from vasculitis should follow to implement targeted interventions to reduce these disparities and to ultimately achieve gender equality in health and healthcare.

In our study, we found that over 90% of the deaths occurred in White patients with vasculitis both as MCD and as the UCD. The mortality rates were slightly different for vasculitis as MCD and as the UCD when stratifying by gender and race. The highest mortality rate for vasculitis as MCD occurred in White patients and White females, and for vasculitis as the UCD the highest rate was in American Indians or Alaska Natives and American Indian or Alaska Native females. Most of the primary systemic vasculitides have race/ethnic differences in prevalence and incidence [26] and it is expected that these differences affect the distribution of mortality. In North America, AAV incidence is greater in White than Black people, which has been postulated to be related to differences in human leukocyte antigen (HLA) [27,28]. GCA has been reported to be more frequent in White people in the United States [29,30] and is rare among non-Caucasian populations in North America, including African Americans [30], Alaska Natives [31], people of Aboriginal descent in Canada [32], and Asians [33]. The highest vasculitis mortality rates in White patients as MCD in our study most likely reflects the fact that GPA and GCA, which were the vasculitides with the highest mortality rates, are more prevalent in White people. The highest age-adjusted mortality rates in American Indians and American Indian females for vasculitis as the UCD most likely reflects disparities in the access to medical care.

We found an exceptionally low mortality rate for vasculitis in Hispanic people. The incidence of primary systemic vasculitides for the Hispanic population is not well known. A study found no differences between the rates of temporal artery biopsy positivity among those who identified themselves as Hispanic [34]. A retrospective study published in 2014 that used the data of two large hospitals in Chicago found that compared to White people, Hispanic patients with AAV present with more severe disease and higher damage indices [35], and this has been corroborated by a more recent report from Loma Linda, California [36]. If this were true, higher vasculitis mortality rates in Hispanic rather than White people could be expected, but this was not the case in our study. In our two previous reports about vasculitis mortality [18,19], we found that the vasculitis mortality rates were very low in Hispanic patients, which was also found in another study on AAV mortality [21]. It is important to keep in mind that these three previous reports used earlier versions of the same database that we used for our current study. Misclassification of Hispanic ethnicity in death certificates could have contributed to our findings. Alternative hypotheses for this differential in mortality rates could be an underdiagnosis of vasculitis, or differences in genetic susceptibility with respect to other ethnic groups. These hypotheses should be analyzed further in subsequent studies.

In our study, we found geographic disparity in the mortality rates for vasculitis both as MCD and as the UCD, with higher rates in northern states. Most of the primary systemic vasculitides have geographic differences in prevalence and incidence, which presumably affects mortality rates. In the United Kingdom and northern Europe, Proteinase 3 antineutrophil cytoplasmic antibodies (PR3-ANCA) and GPA are more common than myeloperoxidase antineutrophil cytoplasmic antibodies (MPO-ANCAs); however, in southern Europe, Asia, and India, MPO-ANCA sand MPA are more common than PR3-ANCA and GPA [26]. In the United States, there is a similar trend, with more PR3-ANCA and GPA in northern states and more MPO-ANCAs and MPA in southern states. Differences in HLA have been postulated to be involved in the geographic and racial differences [27,37,38]. Several studies have reported a higher incidence and prevalence of GCA in Canada [39], in the northern states of the United States [29], as well as in northern Europe [40] than in southern Europe [41] and non-European populations [42]. Therefore, it is not surprising that we found higher mortality rates of vasculitis in northern states, as GPA and GCA were the vasculitis subtypes with the highest mortality in our study. Genetic, environmental and migration patterns may explain, at least in part, this north–south gradient in Europe and the United States.

We found that GPA was the vasculitis responsible for most of the deaths, followed by GCA and anti-GMB disease. For these three vasculitides, the other UCDs, apart from the disease itself, probably reflect complications of the vasculitis or its treatment, such as cardiovascular, stroke, and pulmonary-related deaths. A meta-analysis of mortality in AAV found that the mortality risk was similar to the overall SMR when the analysis was restricted to patients with GPA with a 2.63-fold increase in mortality when compared to the general population [22]. Studies regarding mortality among patients with GCA have yielded conflicting results. Several studies have reported that the life expectancy of patients with GCA does not significantly differ from that of the general population [40,43,44], while another study found that hospitalized GCA patients had lower mortality compared to the general inpatient population [45]. A recent large population-based study in Israel found that patients with GCA had a minor decrease in long-term survival [46]. The reasons why GCA was the vasculitis with the second highest mortality rate in our study are not known, but, in agreement with previous studies, it contributed significantly less than GPA to the overall vasculitis mortality. Anti-GMB disease has been reported to have an overall inpatient prevalence of 10.3 cases per 1,000,000 admissions in the United States with an in-patient mortality rate of 8% [47]. Without prompt diagnosis and treatment, patients with anti-GMB disease can develop organ failure, resulting in significant morbidities and mortality, which may explain why most of the patients with anti-GMB disease in our population-based study died from the disease.

There was a clear trend of a lower vasculitis mortality rate as the size of the population in a specific area increased. All-cause mortality rates in rural areas have exceeded those in urban areas of the United States since the 1980s, and the gap is getting wider. Overall, higher rural mortality may be primarily explained by three factors: socioeconomic deprivation, physician shortages, and lack of health insurance [48]. In the case of rare diseases, as is the case for vasculitis, it is even more obvious that the patients benefit from highly specialized care, which is usually available in large metropolitan areas.

Our study has several limitations. First, to correctly interpret the change in vasculitis mortality and to comment on the differences in mortality according to gender, race, and geographic region, we should know the overall prevalence of vasculitis and each vasculitis type among these groups and geographic regions, as well as the change in the overall mortality rate over time in the abovementioned groups. Unfortunately, this information is not publicly available. Second, as in any similar epidemiological study, there are potential errors related to misclassification in death certificates [49]. Mortality data from death certificates frequently underestimate the burden associated with autoimmune diseases [50], as it is possible that, on some death certificates, the organ-specific cause but not vasculitis may have been recorded. In general, the accuracy and completeness of causes of death in death certification varies in relation to the qualifications of the death certifier, the clinical or diagnostic setting where this certification occurs, as well as the prevalence of a particular cause of death in the population studied. The impact of these variables on the accuracy of vasculitis as a cause of death could not be determined in this study, as this information is not available in the database. Studies specifically designed to address these questions are warranted. The use of the ICD-10 coding system makes comparison with other previous studies, which used the ICD-9 coding system, challenging. In addition, the information included on the death certificate about the race and Hispanic ethnicity of the decedent may be inaccurate because it is reported by the funeral director, as provided by either the surviving next of kin or based on observations. Race and ethnicity information from the census is self-reported. If race and Hispanic origin information is inconsistent between these two sources, death rates will be biased [15]. Our findings should be interpreted with caution until the reliability of the data for rare diseases available in national databases is confirmed, but this also applies to any study with a similar design.

An important contribution of our study is that it found gender, racial, and geographic disparities in the distribution of vasculitis mortality across states and across the urban–rural continuum in the United States. If these differences are confirmed, further studies are warranted to clarify the reasons for this in order to develop targeted action plans to address such disparities.

## 5. Conclusions

There is a clear trend of a progressive decrease in the vasculitis mortality rates in the United States. Increased awareness of these rare diseases, leading to earlier diagnosis and treatment, less toxic therapeutic regimens, and better surveillance and prophylaxis of the long-term complications of both diseases and their treatments, have probably played a role in this progressive decrease in mortality rates. One of the advantages of this study is that the use of population data allowed us to make observations that would have otherwise been difficult to find in small studies, especially for rare diseases. An important contribution of our study is that it found gender, racial, and geographic disparities in the distribution of vasculitis mortality across states and across the urban–rural continuum in the United States.

## Figures and Tables

**Figure 1 jcm-10-01759-f001:**
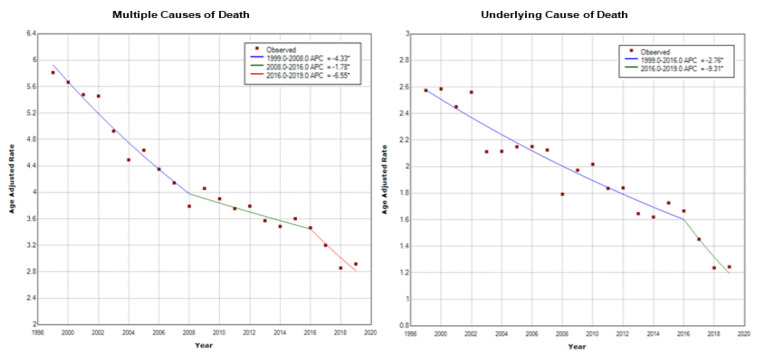
Decreases in the age-adjusted mortality rates for vasculitis as multiple (**left**) and underlying causes of death (**right**) by year, based on joinpoint analysis. APC, annual percent change.

**Figure 2 jcm-10-01759-f002:**
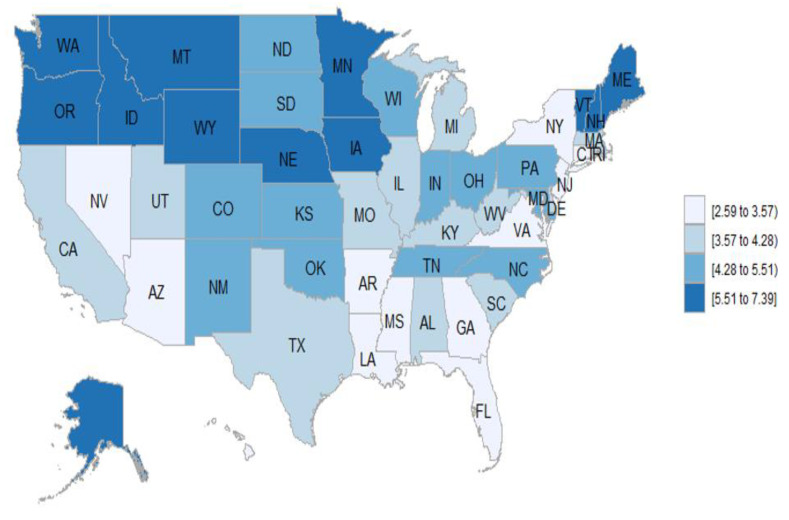
Age-adjusted mortality rates per million for vasculitis as multiple cause of death, by state, from 1999 to 2019.

**Figure 3 jcm-10-01759-f003:**
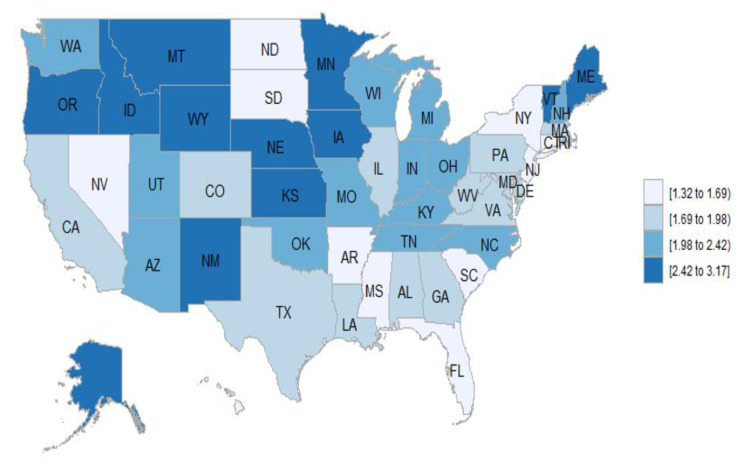
Age-adjusted mortality rates per million for vasculitis as the underlying cause of death, by state, from 1999 to 2019.

**Table 1 jcm-10-01759-t001:** Age-adjusted mortality rates per million for vasculitis as multiple causes of death and as underlying cause of death, by race and gender.

		Multiple Causes of Death			Underlying Cause of Death		
**Race**	**Gender**	**Deaths**	**Age-Adjusted Rate (95% CI)**	**Percentage of Total Deaths**	**Deaths**	**Age-Adjusted Rate (95% CI)**	**Percentage of Total Deaths**
White	Female	15,256	4.469 (4.397–4.542)	54.52%	6622	2.006 (1.957–2.056)	50.75%
	Male	10,583	4.176 (4.095–4.256)	37.82%	5298	2.052 (1.996–2.108)	40.60%
	**Total**	**25,839**	**4.371** **(4.317–4.424)**	**92.34%**	**11,920**	**2.032** **(1.995–2.069)**	**91.36%**
American Indian or Alaska Native	Female	102	3.582 (2.851–4.313)	0.37%	70	2.429 (1.867–3.108)	0.54%
	Male	80	3.275 (2.516–4.190)	0.29%	40	1.594 (1.090–2.250)	0.31%
	**Total**	**182**	**3.451** **(2.910–3.992)**	**0.65%**	**110**	**2.059** **(1.643–2.475)**	**0.84%**
Black or African American	Female	843	2.088 (1.946–2.231)	3.013%	397	0.978 (0.881–1.076)	3.04%
	Male	599	1.970 (1.800–2.140)	2.14%	348	1.099 (0.975–1.223)	2.67%
	**Total**	**1442**	**2.037** **(1.929–2.146)**	**5.15%**	**745**	**1.024** **(0.948–1.099)**	**5.71%**
Asian or Pacific Islander	Female	295	1.892 (1.672–2.111)	1.05%	152	0.944 (0.791–1.097)	1.17%
	Male	225	1.795 (1.548–2.041)	0.80%	121	0.944 (0.768–1.120)	0.93%
	**Total**	**520**	**1.861** **(1.696–2.026)**	**1.86%**	**273**	**0.941** **(0.826–1.057)**	**2.09%**
Total		**27,983**	**4.077** **(4.029–4.125)**	**100.00%**	**13,048**	**1.888** **(1.855–1.921)**	**100.00%**

CI, confidence interval. Partial and overall total are in bold.

**Table 2 jcm-10-01759-t002:** Age-adjusted mortality rates per million, by the specific type of vasculitis (1999–2019), presented in descending order by underlying cause of death.

	Multiple Causes of Death		Underlying Cause of Death	
Type of Vasculitis	Deaths	Age-Adjusted Rate (95% CI)	Deaths	Age-Adjusted Rate (95% CI)
Granulomatosis with polyangiitis (GPA)	10,388	1.514 (1.485–1.544)	6547	0.962 (0.939–0.986)
Giant cell arteritis (GCA)	9200	1.329 (1.302–1.356)	1823	0.268 (0.256–0.281)
Antiglomerular basement membrane (anti-GBM) disease	1910	0.291 (0.277–0.304)	1314	0.197 (0.186–0.209)
Polyarteritis nodosa (PAN)	1694	0.239 (0.227–0.251)	845	0.126 (0.117–0.135)
Eosinophilic granulomatosis with polyangiitis (EGPA)	1261	0.178 (0.168–0.189)	681	0.099 (0.091–0.107)
Cryoglobulinemia	1353	0.195 (0.184–0.206)	512	0.068 (0.062–0.074)
Microscopic polyangiitis (MPA)	742	0.093 (0.086–0.100)	508	0.063 (0.057–0.069)
Takayasu arteritis (TAK)	533	0.079 (0.071–0.086)	289	0.035 (0.030–0.040)
Behcet’s disease	519	0.071 (0.064–0.078)	252	0.033 (0.028–0.038)
Ig (immunoglobulin) A vasculitis (Henoch–Schönlein) (IgAV)	384	0.042 (0.037–0.046)	166	0.014 (0.011–0.017)
Kawasaki disease	177	0.013 (0.010–0.016)	102	0.008 (0.006–0.011)
Other conditions related to polyarteritis nodosa	16	Unreliable	6	Unreliable
Giant cell arteritis with polymyalgia Rheumatica	12	Unreliable	2	Unreliable
Juvenile polyarteritis	1	Unreliable	1	Unreliable
**Total**	**28,190**	**4.096** **(4.048–4.144)**	**13,048**	**1.888** **(1.855–1.921)**

CI, confidence interval. Overall totals are in bold. Underlying Cause of Death (UCD) for the Vasculitides Contributing the Most to the Multiple Cause of Death (MCD) Data.

**Table 3 jcm-10-01759-t003:** Age-adjusted mortality rates per million for vasculitis as the underlying cause of death and multiple causes of death, by race and gender.

	Multiple Causes of Death			Underlying Cause of Death		
2013 Urbanization *	Deaths	Age-Adjusted Rate (95% CI)	Percentage of Total Deaths	Deaths	Age-Adjusted Rate (95% CI)	Percentage of Total Deaths
Micropolitan	3357	5.001 (4.831–5.172)	12.00%	1569	2.350 (2.232–2.468)	12.03%
Small metro	3321	4.930 (4.761–5.099)	11.87%	1525	2.265 (2.150–2.380)	11.67%
Noncore	2398	4.546 (4.362–4.731)	8.57%	1105	2.134 (2.006–2.262)	8.47%
Medium metro	6234	4.294 (4.186–4.401)	22.28%	2890	2.017 (1.943–2.092)	22.15%
Large fringe metro	6140	3.762 (3.667–3.857)	21.94%	2869	1.742 (1.677–1.806)	21.99%
Large central metro	6533	3.437 (3.353–3.521)	23.35%	3090	1.611 (1.554–1.668)	23.68%
Total	27,983	4.077 (4.029–4.125)	100%	13,048	1.888 (1.855–1.921)	100%

* 2013 Urbanization refers to the National Center for Health Statistics (NCHS) Urban–Rural Classification Scheme for Counties. Metropolitan counties: Large central metro counties in metropolitan statistical areas (MSAs) with populations of 1 million that (1) contain the entire population of the largest principal city of the MSAs, (2) are completely contained within the largest principal city of the MSA, or (3) contain at least 250,000 residents of any principal city in the MSAs. Large fringe metro counties in MSAs with a population of 1 million or more that do not qualify as large central medium metro counties in MSAs with a population of 250,000–999,999. Small metro counties are counties in MSAs with a population of less than 250,000. Micropolitan relates to an urban area with a population of at least 10,000 but less than 50,000. Nonmetropolitan counties: Micropolitan counties in micropolitan statistical areas; noncore counties not in micropolitan statistical areas.

## Data Availability

Data are available in a publicly accessible database: CDC Wonder.

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
