# Peer review of "Trend and Geographic Disparities in the Mortality Rates of Primary Systemic Vasculitis in the United States from 1999 to 2019: A Population-Based Study"

_jcm, 2021, doi:10.3390/jcm10081759_

Round 1

Reviewer 1 Report

Dear Authors,

first of all, I thank You for giving me the opportunity to read this Your manuscript.

Here are my comments and suggestions. I hope that they are useful. for You.

1) Introduction: it seemed appropriate to write about the classification of primary systemic vasculitis. You should not take for granted concepts that may be unfamiliar to many readers. Please, re-write Introduction section according to this suggestion.

2) The sentence "The geographic distribution of mortality......to the geographic distribution of overall mortality" should be removed because it was redundant and not useful.    

3) Discussion: some points were only mentioned but not really discussed. For instance, You found an exceptionally low mortality rate for vasculitis in Hispanic people: this depended only on misclassification of Hispanic ethnicity in death certificates ?  It was unclear.

Besides, You found higher mortality rates for vasculitis in northern countries, and highlighted the existence of a similar trend regarding PR3-ANCA, GPA, MPO-ANCA, MPA in northern and southern Europe. Please, discuss this point. 

Finally, the role of the diagnostic setting was not properly evaluated. In short, can a death certificate written in immunology/rheumatology have the same "value" as a death certificate written in non-specialist setting ?   

Author Response

Point 1: Introduction: it seemed appropriate to write about the classification of primary systemic vasculitis. You should not take for granted concepts that may be unfamiliar to many readers. Please, re-write Introduction section according to this suggestion. 

Response 1: We thank the reviewer for the insightful review of our manuscript and his/her comments. In response to his/her first comment, we have re-written the Introduction to include an explanation about the classification of the primary systemic vasculitides. We have added the following in line 28 of page 1: “Large vessel vasculitis primarily affects the aorta and supraaortic branches. Giant cell arteritis (GCA) and Takayasu arteritis (TAK) are the two major variants of large vessel vasculitis. Medium vessel vasculitis primarily affects the main visceral arteries and their branches. Polyarteritis nodosa (PAN) and Kawasaki disease are the major variants. Small vessel vasculitis predominantly affects the intraparenchymal arteries, arterioles, capillaries, and venules. The two major categories of small vessel vasculitis are the subgroup of anti-neutrophil cytoplasmic antibodies (ANCA)-associated vasculitis (AAV), which includes granulomatosis with polyangiitis (GPA), eosinophilic granulomatosis with polyangiitis (EGPA), and microscopic polyangiitis (MPA), and the immune complex small vessel vasculitis, which includes anti-glomerular basement membrane (anti-GBM) disease, cryoglobulinemic vasculitis, IgA vasculitis (IgAV) and hypocomplementemic urticarial vasculitis. Additional categories are variable vessel vasculitis, which includes Behcet’s disease; single-organ vasculitis; and vasculitis associated with systemic disease or probable etiology [1].”

Point 2: The sentence "The geographic distribution of mortality......to the geographic distribution of overall mortality" should be removed because it was redundant and not useful.

Response 2: We thank the reviewer for his comment. In response to this comment, we have deleted from line 18 of page 2: “to the geographic distribution of overall mortality.”

Point 3: Discussion: some points were only mentioned but not really discussed. For instance, You found an exceptionally low mortality rate for vasculitis in Hispanic people: this depended only on misclassification of Hispanic ethnicity in death certificates?  It was unclear.

Response 3: We thank the reviewer for this comment. In response to this critique, we have added the following in line 20 of page 9: “Alternative hypotheses for this differential in mortality rates could be an underdiagnosis of vasculitis, or differences in genetic susceptibility with respect to other populations. These hypotheses should be analyzed further in subsequent studies.”

Point 4: Besides, You found higher mortality rates for vasculitis in northern countries, and highlighted the existence of a similar trend regarding PR3-ANCA, GPA, MPO-ANCA, MPA in northern and southern Europe. Please, discuss this point.

Response 4: We are grateful to the reviewer for this comment, as well. Following his/her suggestion, we have added the following sentence in line 38 of page 9: “Genetic, environmental and migration patterns may explain, at least in part, this north-south gradient in Europe and the United States.”

Point 5: Finally, the role of the diagnostic setting was not properly evaluated. In short, can a death certificate written in immunology/rheumatology have the same "value" as a death certificate written in non-specialist setting?

Response 5: In response to this comment, we have added the following explanation in line 23 of page 10, which we think enhances the discussion of the limitations of our study: “In general, the accuracy and completeness of causes of death in death certification varies in relation to the qualifications of the death certifier, the clinical or diagnostic setting where this certification occurs, as well as the prevalence of a particular cause of death in the population studied. The impact of these variables on the accuracy of vasculitis as a cause of death could not be determined in this study, because this information is not available in the database. Studies specifically designed to address these questions are warranted.”

Again, we are incredibly grateful to the reviewer for the insightful comments.

Reviewer 2 Report

This is a population-based study investigating mortality rates for vasculitis in the United States utilizing a CDC database for the years 1999 to 2019. The authors describe that mortality from vasculitis has been decreasing overall, but there is a marked north-south divide in mortality rates and vastly differeing mortality in the various ethnic groups, with Hispanics having the lowest mortality. This reviewer would like to congratulate the authors on a well conducted study on a quite interesting subject.

There are no major problems with this study; the data is very well researched and presented. The term 'micropolitan' could maybe be explained to the uninitiated reader – it was news to this reviewer. This reviewer has no objection to the publication of this paper.

Author Response

Point 1: This is a population-based study investigating mortality rates for vasculitis in the United States utilizing a CDC database for the years 1999 to 2019. The authors describe that mortality from vasculitis has been decreasing overall, but there is a marked north-south divide in mortality rates and vastly differing mortality in the various ethnic groups, with Hispanics having the lowest mortality. This reviewer would like to congratulate the authors on a well conducted study on a quite interesting subject.

Response 1: The authors would like to deeply thank the reviewer for his/her insightful review and nice comments about our work.

Point 2: There are no major problems with this study; the data is very well researched and presented. The term 'micropolitan' could maybe be explained to the uninitiated reader – it was news to this reviewer. This reviewer has no objection to the publication of this paper.

Response 2: We thank the reviewer for raising awareness about this point. In response to this comment, we have added the following to the footnote of Table 3: “Micropolitan relates to an urban area with a population of at least 10,000 but less than 50,000.”

Round 2

Reviewer 1 Report

Dear Authors,

I read the newer, revised version of Your manuscript.

All my comments and suggestions were met. However, You should have made explicit reference to the Chapel Hill classification. Introduction should be revised according to this suggestion.

Author Response

Point 1: All my comments and suggestions were met. However, You should have made explicit reference to the Chapel Hill classification. Introduction should be revised according to this suggestion.

Response 1: We thank the reviewer for this comment. In response to it, we have added the following sentence to the introduction: “According to the 2012 International Chapel Hill Consensus Conference on the Nomemclature of Vasculitides,” The change is on page 1, line 27, and has been written in blue.